# A stochastic gradient descent algorithm with random search directions

**Eméric Gbaguidi**                                    *thierry-emeric.gbaguidi@math.u-bordeaux.fr*
*Institut de Mathématiques de Bordeaux*
*Université de Bordeaux*

**Reviewed on OpenReview:** *https://openreview.net/forum?id=npER8AaLSb*

## Abstract

Stochastic coordinate descent algorithms are efficient methods in which each iterate is obtained by fixing most coordinates at their values from the current iteration, and approximately minimizing the objective with respect to the remaining coordinates. However, this approach is usually restricted to canonical basis vectors of $\mathbb{R}^d$. In this paper, we develop the class of stochastic gradient descent algorithms with random search directions. These methods use the directional derivative of the gradient estimate following more general random vectors. We establish the almost sure convergence of these algorithms with decreasing step. We further investigate their central limit theorem and pay particular attention to analyze the impact of the search distributions on the asymptotic covariance matrix. We also provide non-asymptotic $\mathbb{L}^p$ rates of convergence.

## 1 Introduction

Consider the unconstrained optimization problem in $\mathbb{R}^d$ which can be written as

$$\min_{x \in \mathbb{R}^d} f(x), \tag{$\mathcal{P}$}$$

where $f$ is the average of many functions,

$$f(x) = \frac{1}{N} \sum_{k=1}^{N} f_k(x). \tag{1}$$

Many computational problems in various disciplines can be formulated as above and the stochastic algorithms are now a common approach to solve ($\mathcal{P}$) since the seminal works of Robbins & Monro (1951) and Kiefer & Wolfowitz (1952). The most famous of them is the standard Stochastic Gradient Descent (SGD) algorithm given, for all $n \geqslant 1$, by

$$X_{n+1} = X_n - \gamma_n \nabla f_{U_{n+1}}(X_n), \tag{SGD}$$

where the initial state $X_1$ is a squared integrable random vector of $\mathbb{R}^d$ which can be arbitrarily chosen, $\nabla f(X_n)$ is the gradient of the function $f$ calculated at the value $X_n$, $(U_n)$ is a sequence of independent and identically distributed random variables with uniform distribution on $\{1, 2, \ldots, N\}$, which is also independent from the sequence $(X_n)$. Moreover, $(\gamma_n)$ is a positive deterministic sequence decreasing towards zero and satisfying the standard conditions

$$\sum_{n=1}^{\infty} \gamma_n = +\infty \qquad \text{and} \qquad \sum_{n=1}^{\infty} \gamma_n^2 < +\infty. \tag{2}$$

Despite its computational efficiency provided by using a random gradient estimate, the SGD algorithm still requires the computation of a vector of size $d$ at each iteration. However, in large-scale machine

learning problems, the reduction of the calculation cost remains one of the main challenges. Moreover, the vast majority of SGD algorithms update all coordinates in the same way (Robbins & Monro, 1951; Gower et al., 2021b; Leluc & Portier, 2022). These issues led to the development of Stochastic Coordinate Gradient Descent (SCGD) algorithms which are very easy to implement and have become unavoidable in high-dimensional framework (Nesterov, 2012; Shalev-Shwartz & Zhang, 2013; Wright, 2015). This class of methods have received a great deal of attention in recent years due to their potential for solving large-scale optimization problems (Lin et al., 2014; Leluc & Portier, 2022).

The SCGD algorithm modifies the SGD algorithm in the sense that its update rule is given, for all $n \geqslant 1$, by

$$
\begin{cases}
X_{n+1}^{(j)} = X_n^{(j)} & \text{if } j \neq \xi_{n+1}, \\
X_{n+1}^{(j)} = X_n^{(j)} - \gamma_n g_{n+1}^{(j)} & \text{if } j = \xi_{n+1},
\end{cases}
\tag{3}
$$

where $X_n^{(j)}$ stands for the $j$-th component of a vector $X_n$, $g_{n+1} = \nabla f_{U_{n+1}}(X_n)$ and $(\xi_n)$ is a sequence of random variables with values in $\{1, 2, \ldots, d\}$ used to select a coordinate of the gradient estimate and follows distribution called the *coordinate sampling policy*.

Hence, the SCGD algorithm selects and updates one coordinate at each iteration in order to sufficiently reduce the objective value while keeping other coordinates fixed. In fact, this approach can also be seen as the SGD algorithm applied just on one random coordinate. There are many strategies for the choice of the coordinate sampling policy. Our goal is to go further in the analysis of the SCGD algorithms with decreasing step. Firstly, we develop the class of stochastic gradient descent algorithms with random search directions. These methods include the SCGD algorithms and use the directional derivative of the gradient estimate following more general random vectors. Thus, beyond the SCGD algorithms, we consider the random directions sampled from gaussian and spherical distributions (see Section 3). Based on weak hypotheses associated to the objective function, we establish the almost sure convergence of these algorithms with decreasing step (see Section 4.1). Secondly, we investigate their central limit theorem and we obtain that the asymptotic covariance matrix depends on the choice of the search direction distributions (see Section 4.2). Therefore, we provide an introductory analysis on the asymptotic performances for different choices of random directions. Lastly, we provide non-asymptotic $\mathbb{L}^p$ rates of convergence for the SGD algorithms with random search directions (see Section 4.3).

**Organization of the paper.**

The rest of the paper is organized as follows. Section 2 summarizes the state of art on the SCGD algorithms. In Section 3, we introduce the theoretical framework of our paper. Section 4 is devoted to our main results. Finally, in Section 5 we illustrate the good performances of the algorithms with decreasing step through numerical experiments on simulated data. All technical proofs are postponed in appendices.

## 2 Related work

Several stochastic coordinate descent algorithms and their variants were proposed and analyzed over years in (Shalev-Shwartz & Tewari, 2009; Nesterov, 2012; Richtárik & Takáč, 2016; Gorbunov et al., 2020; Leluc & Portier, 2022; Ramesh et al., 2023). They are recursive stochastic algorithms in which each iterate is obtained by fixing most coordinates at their values from the current iteration, and approximately minimizing the objective with respect to the remaining coordinates (Wright, 2015). For Nesterov (2012), the main advantage of the coordinate descent methods is the simplicity of each iteration, both in generating the descent direction and in updating of the variables. Hence, many questions have been addressed in the literature on these algorithms due to their considerable interest.

On the one hand, the choice of the coordinate sampling policy is clearly a major issue. From that, we can distinguish two main classes of coordinate selection rules: *deterministic and stochastic.* However, Sun et al. (2017) considered that the stochastic block rule is easier to analyze because taking expectation will yield a good approximation to the full gradient and ensures that every coordinate is updated at the specified frequency. Richtárik & Takáč (2016) also obtained that the scheme of updating a single randomly selected

coordinate per iteration with optimal probabilities may require less iterations to converge, than all coordinates updating rule at every iteration. For its part, Needell et al. (2014) suggested a non-uniform sampling for the stochastic gradient descent. Moreover, (Chang et al., 2008; Shalev-Shwartz & Zhang, 2013) proved that the randomized strategies are more efficient than the simple rule of cycling through the coordinates (Luo & Tseng, 1992; Beck & Tetruashvili, 2013; Saha & Tewari, 2013). Nevertheless, Nutini et al. (2015) compared and showed that the deterministic Gauss-Southwell rule is faster than the random coordinate selection rule in some empirical cases. However, Saha & Tewari (2013) remember us that Gauss–Southwell rule typically takes $\mathcal{O}(d)$ time to implement instead of just $\mathcal{O}(1)$ for the uniform randomized rule. This was the main motivation for the choice of uniform probabilities in Shalev-Shwartz & Tewari (2009) which can also justify a good performance of the SCGD algorithms.

On the other hand, Tao et al. (2012) proposed and showed the convergence rates of stochastic coordinate descent methods adapted for the regularized smooth and non-smooth losses. Likewise, Beck & Tetruashvili (2013) established a global sublinear rate of convergence of the block coordinate gradient projection method with constant stepsize which depends on the Lipschitz parameters. Later, Lin et al. (2014) developed an accelerated randomized proximal coordinate gradient method which achieves faster linear convergence rates for minimizing strongly convex functions than existing randomized proximal coordinate gradient methods. Moreover, Konečnỳ et al. (2017) introduced the semi-stochastic coordinate descent algorithm with constant stepsize which picks a random function and a random coordinate both using non-uniform distributions at each step. They proved the convergence of $f(X_n)$ towards the minimum of $f$ in $\mathbb{L}^1$ under the strong convexity assumption on $f$. There exist many others contributions on the stochastic coordinate descent variants. For instance, Richtárik & Takáč (2016) investigated the parallel coordinate descent method in which a random subset of coordinates is chosen and updated at each iteration. Allen-Zhu et al. (2016) also studied the accelerated coordinate descent algorithm using non-uniform sampling. Kozak et al. (2019) and Hanzely et al. (2018) studied the projected gradient algorithms onto a random subspace at each iteration with constant stepsize. Hanzely et al. (2018) established the convergence only in expectation by assuming that $f$ is $\mu$-strongly convex. We also highlight that Kozak et al. (2019) proved the almost sure convergence of $(X_n)$ under the same restrictive assumption of strong-convexity. However, both of these papers compute and use the full gradient $\nabla f(X_n)$ at each iteration. This will not be the case at all here, since we use a uniformly chosen random estimate of the gradient. Moreover, in our paper, we consider a decreasing stepsize $\gamma_n$ to avoid dependence on $\mu$-strong convexity and $L$-smoothness constants.

More recently, Leluc & Portier (2022) provided the almost sure convergence as well as non-asymptotic bounds of the SCGD algorithm with decreasing step in a non-convex setting and including zeroth-order gradient estimate. They proved the convergence of the SCGD iterates towards stationary points in the sense that $\nabla f(X_n)$ converges to 0 almost surely. They also proposed non-asymptotic bounds on the optimality gap $\mathbb{E}[f(X_n) - f^*]$ where $f^*$ is a lower bound of $f$. These results have been established by assuming that $f$ is $L$-smooth and under the growth condition (Leluc & Portier, 2022; Gower et al., 2021b). The non-asymptotic bounds required the additional Polyak–Lojasiewicz (PL) condition (Polyak, 1963; Gower et al., 2021a).

Although the assumptions used in Leluc & Portier (2022) are relatively weak, one can still try to relax them and provide the algorithm convergence directly on the sequence $(X_n)$. Furthermore, one can also be interested by non-asymptotic $\mathbb{L}^p$ rates of convergence for any integer $p$. To our best knowledge, the central limit theorem for the SCGD algorithm was not previously established.

## 3   Preliminaries

In this section, we present the mathematical background of the paper. Firstly, we introduce some notations. Secondly, we shall formulate our class of SGD algorithms with random search directions which includes the SCGD algorithm. Finally, we will spell out some regularity assumptions. In order to avoid any ambiguity, every vector norm is the $\ell_2$ norm.

The SCGD algorithm as given in (3), represents the practical definition by considering the vectors in the canonical basis of $\mathbb{R}^d$. However, we can extend this coordinate selecting rule by using more general random vectors with a possible adaptive sampling policy. Therefore, we introduce the Stochastic Coordinate Gradient

Descent algorithm with Random Search Direction (SCORS) defined for all $n \geqslant 1$, by

$$X_{n+1} = X_n - \gamma_n D(V_{n+1}) \nabla f_{U_{n+1}}(X_n), \qquad \text{(SCORS)}$$

where the initial state $X_1$ is a squared integrable random vector of $\mathbb{R}^d$ which can be arbitrarily chosen, $D(v) = vv^T$ for any vector $v \in \mathbb{R}^d$, the sequence $(U_n)$ is independent from $(X_n)$ where $(U_n)$ is independent and identically distributed with $\mathcal{U}(\llbracket 1, N \rrbracket)$ distribution and $V_n$ is a random vector of $\mathbb{R}^d$ sampled from an underlying distribution $\mathcal{P}_n$ satisfying certain conditions (see Assumption 1 below). Furthermore, we assume that $V_{n+1}$ is independent from $U_{n+1}$ conditionally on $\mathcal{F}_n$, where $\mathcal{F}_n = \sigma(X_1, \ldots, X_n)$ is the $\sigma$-field associated to the sequence of iterates $(X_n)$. In the same way as before, $(\gamma_n)$ is a positive deterministic sequence decreasing towards zero and satisfying the standard conditions (2).

Moreover, we will consider the random direction $V_{n+1}$ that satisfies the distributional constraint $\mathbb{E}[D(V_{n+1})|\mathcal{F}_n] = \mathbf{I}_d$ in order to reduce the bias in the SCORS gradient estimate.

**Assumption 1.** *For all $n \geqslant 1$, we assume that the random direction vector $V_n$ is sampled from an independent and possibly adaptive distribution $\mathcal{P}_n$ such that*

$$\mathbb{E}[D(V_{n+1})|\mathcal{F}_n] = \mathbf{I}_d \qquad a.s. \qquad (4)$$

*In addition, we suppose that the 4-th conditional moment of $V_{n+1}$ is bounded, i.e. there exists a positive constant $m_4$ such that $\mathbb{E}[\|V_{n+1}\|^4|\mathcal{F}_n] \leqslant m_4$ for all $n \geqslant 1$.*

**Choices of the direction distribution.** We conduct here a comprehensive analysis on different choices of distributions $\mathcal{P}_n$ satisfying Assumption 1. Several popular choices were proposed in (Chen et al., 2024) and are listed as follows.

(**U**) Uniform in the canonical basis: $V_n$ is sampled from $\{\sqrt{d}e_1, \ldots, \sqrt{d}e_d\}$ with equal probability $1/d$, where $(e_1, \ldots, e_d)$ is the canonical basis of $\mathbb{R}^d$.

(**NU**) Non-uniform in the canonical basis with probabilities $(p_{n,1}, \ldots, p_{n,d})$: $V_n$ is sampled such that for all $n \geqslant 1$ and $j \in \llbracket 1, d \rrbracket$,

$$V_n = \sqrt{\frac{1}{p_{n,\xi_n}}} e_{\xi_n}, \qquad (5)$$

where $(\xi_n)$ is a sequence of independent random variables defined on $\llbracket 1, d \rrbracket$ such that

$$\mathbb{P}[\xi_n = j|\mathcal{F}_{n-1}] = p_{n,j}, \qquad (6)$$

with $p_{n,j} > 0$ for all $j \in \llbracket 1, d \rrbracket$ and $\sum_{j=1}^d p_{n,j} = 1$.

(**G**) Gaussian: $(V_n)$ is sampled from the normal distribution $\mathcal{N}(0, \mathbf{I}_d)$.

(**S**) Spherical: we sample $(V_n)$ from the uniform distribution on the sphere with Euclidean norm $\|V_n\|^2 = d$.

It is easy to see that in all above choices, the search distribution $\mathcal{P}_n$ satisfies the equality (4). Moreover, in the particular uniform sampling case, the random coordinate is uniformly chosen at each iteration. In others words, the coordinate along which the descent shall proceed, is selected in $\llbracket 1, d \rrbracket$ with the same probability equal to $1/d$. Furthermore, we can consider many procedures to compute the probabilities $(p_{n,1}, \ldots, p_{n,d})$ in the non-uniform case. We will spell out different strategies in the numerical experiments.

However, one can remark that the (**U**) and (**NU**) direction distribution choices correspond to the standard SCGD algorithms (3). Thus, the most interesting contribution here lies in the cases (**G**) and (**S**) by taking into account random directions of descent other than those of the canonical basis vectors of $\mathbb{R}^d$. The SCORS algorithm can also be interpreted as the use of the directional derivative of the gradient estimate following given random vectors. Malladi et al. (2023) explored a memory-efficient zeroth-order algorithm with projected gradient for fine-tuning of large language models. The directional derivative can be computed in forward-mode and this requires far less memory than backward passes. Therefore, the SCORS algorithm proposed here, may also find an interest for such applications in fine-tuning. In the sequel, we state the others general assumptions required for our analysis and used throughout the paper.

**Assumption 2.** *Assume that the function $f$ is continuously differentiable with a unique equilibrium point $x^*$ in $\mathbb{R}^d$ such that $\nabla f(x^*) = 0$.*

**Assumption 3.** *Suppose that for all $x \in \mathbb{R}^d$ with $x \neq x^*$,*

$$\langle x - x^*, \nabla f(x) \rangle > 0.$$

**Assumption 4.** *Assume there exists a positive constant $L$ such that, for all $x \in \mathbb{R}^d$,*

$$\frac{1}{N} \sum_{k=1}^{N} \|\nabla f_k(x) - \nabla f_k(x^*)\|^2 \leqslant L\|x - x^*\|^2.$$

Assumption 2 is a standard hypothesis for the study of SGD algorithms. Furthermore, Assumption 3 does not impose a restrictive strong convexity hypothesis on $f$. We need to note though that this condition is much weaker than saying that $f$ is strictly convex. Moreover, our Assumption 4 is of key importance and essential in general non-convex setting. This condition states that at the optimal point, the gradient of all functions $f_k$ for any $1 \leqslant k \leqslant N$, does not change arbitrarily with respect to the vector $x \in \mathbb{R}^d$. Let us link our assumption with a Lipschitz condition for the gradient of the objective function. It is obvious that if each function $f_k$ has Lipschitz continuous gradient with constant $\sqrt{L_k}$, then Assumption 4 is satisfied by taking $L$ as the average value of all $L_k$. Assumption 4 is also less restrictive than the growth conditions with the $\sqrt{L}$-smoothness of $f$, which are classical assumptions among the literature. We point out that both Assumptions 3 and 4 are enough suitable to obtain the almost sure convergence of the SCORS algorithm.

## 4 Main results

We first present the almost sure convergence of the SCORS algorithm. Then, we focus on the central limit theorem for the SCORS algorithm, and finally we propose non-asymptotic $\mathbb{L}^p$ rates of convergence.

### 4.1 Almost sure convergence

**Theorem 1.** *Consider that $(X_n)$ is the sequence generated by the SCORS algorithm with decreasing step sequence $(\gamma_n)$ satisfying (2). In addition, suppose that Assumptions 1, 2, 3 and 4 are satisfied. Then, we have*

$$\lim_{n \to +\infty} X_n = x^* \qquad a.s., \tag{7}$$

*and*

$$\lim_{n \to +\infty} f(X_n) = f(x^*) \qquad a.s. \tag{8}$$

*Proof.* Recall that for all $n \geqslant 1$,

$$X_{n+1} = X_n - \gamma_n D(V_{n+1}) \nabla f_{U_{n+1}}(X_n). \tag{9}$$

Let us consider the Lyapunov function $T_n$ defined for all $n \geqslant 1$, by

$$T_n = \|X_n - x^*\|^2.$$

Hence, it follows that almost surely,

$$\begin{aligned}
T_{n+1} &= \|X_{n+1} - x^*\|^2 \\
&= \|X_n - x^* - \gamma_n D(V_{n+1}) \nabla f_{U_{n+1}}(X_n)\|^2 \\
&= T_n - 2\gamma_n \langle X_n - x^*, D(V_{n+1}) \nabla f_{U_{n+1}}(X_n) \rangle + \gamma_n^2 \|D(V_{n+1}) \nabla f_{U_{n+1}}(X_n)\|^2. 
\end{aligned} \tag{10}$$

However, we have from Assumption 1 that almost surely

$$\mathbb{E}[D(V_{n+1})|\mathcal{F}_n] = \mathbf{I}_d. \tag{11}$$

Moreover, we obtain from (1) and the fact that $U_{n+1}$ is uniformly distributed on $[\![1, N]\!]$ that

$$\mathbb{E}[\nabla f_{U_{n+1}}(X_n)|\mathcal{F}_n] = \nabla f(X_n) \qquad a.s. \tag{12}$$

By putting together (11), (12) and the conditional independence of $V_{n+1}$ and $U_{n+1}$, it immediately follows that

$$\mathbb{E}[D(V_{n+1})\nabla f_{U_{n+1}}(X_n)|\mathcal{F}_n] = \nabla f(X_n) \qquad a.s. \tag{13}$$

Furthermore, we recall by definition that,

$$D(V_{n+1}) = V_{n+1}V_{n+1}^T, \tag{14}$$

which implies that

$$\begin{aligned}
\|D(V_{n+1})\nabla f_{U_{n+1}}(X_n)\|^2 &= \|V_{n+1}V_{n+1}^T \nabla f_{U_{n+1}}(X_n)\|^2 \\
&= \left(\langle V_{n+1}, \nabla f_{U_{n+1}}(X_n)\rangle\right)^2 \|V_{n+1}\|^2.
\end{aligned} \tag{15}$$

By using the Cauchy-Schwarz inequality, we deduce from (15) that

$$\|D(V_{n+1})\nabla f_{U_{n+1}}(X_n)\|^2 \leqslant \|V_{n+1}\|^4 \|\nabla f_{U_{n+1}}(X_n)\|^2. \tag{16}$$

Once again, from the conditional independence of $V_{n+1}$ and $U_{n+1}$ combined with the inequality (16), we obtain that almost surely

$$\mathbb{E}[\|D(V_{n+1})\nabla f_{U_{n+1}}(X_n)\|^2|\mathcal{F}_n] \leqslant \mathbb{E}[\|V_{n+1}\|^4|\mathcal{F}_n]\mathbb{E}[\|\nabla f_{U_{n+1}}(X_n)\|^2|\mathcal{F}_n]. \tag{17}$$

It implies via Assumption 1 that there exists a positive constant $m_4$ such that

$$\mathbb{E}[\|D(V_{n+1})\nabla f_{U_{n+1}}(X_n)\|^2|\mathcal{F}_n] \leqslant m_4 \mathbb{E}[\|\nabla f_{U_{n+1}}(X_n)\|^2|\mathcal{F}_n]. \tag{18}$$

However, we have that almost surely

$$\|\nabla f_{U_{n+1}}(X_n)\|^2 \leqslant 2\Big(\|\nabla f_{U_{n+1}}(X_n) - \nabla f_{U_{n+1}}(x^*)\|^2 + \|\nabla f_{U_{n+1}}(x^*)\|^2\Big). \tag{19}$$

Define for all $x \in \mathbb{R}^d$,

$$\tau^2(x) = \frac{1}{N}\sum_{k=1}^{N}\|\nabla f_k(x) - \nabla f_k(x^*)\|^2.$$

As $U_{n+1}$ is uniformly distributed on $[\![1, N]\!]$, we clearly have

$$\mathbb{E}[\|\nabla f_{U_{n+1}}(X_n) - \nabla f_{U_{n+1}}(x^*)\|^2|\mathcal{F}_n] = \tau^2(X_n) \qquad a.s., \tag{20}$$

and

$$\mathbb{E}[\|\nabla f_{U_{n+1}}(x^*)\|^2|\mathcal{F}_n] = \frac{1}{N}\sum_{k=1}^{N}\|\nabla f_k(x^*)\|^2 \qquad a.s. \tag{21}$$

Then, by using the four inequalities (18), (19), (20) and (21), we obtain that

$$\mathbb{E}[\|D(V_{n+1})\nabla f_{U_{n+1}}(X_n)\|^2|\mathcal{F}_n] \leqslant 2m_4(\tau^2(X_n) + \theta^*) \qquad a.s., \tag{22}$$

where

$$\theta^* = \frac{1}{N}\sum_{k=1}^{N}\|\nabla f_k(x^*)\|^2.$$

Furthermore, from the three contributions (10), (13) and (22), one deduces that

$$\mathbb{E}[T_{n+1}|\mathcal{F}_n] \leqslant T_n - 2\gamma_n\langle X_n - x^*, \nabla f(X_n)\rangle + 2m_4\gamma_n^2(\tau^2(X_n) + \theta^*) \qquad a.s. \tag{23}$$

However, Assumption 4 implies that $\tau^2(X_n) \leqslant LT_n$ almost surely. Consequently, we obtain from (23) that

$$\mathbb{E}[T_{n+1}|\mathcal{F}_n] \leqslant \left(1 + 2Lm_4\gamma_n^2\right)T_n - 2\gamma_n\langle X_n - x^*, \nabla f(X_n)\rangle + 2m_4\theta^*\gamma_n^2 \qquad a.s., \tag{24}$$

which can be rewritten as

$$\mathbb{E}[T_{n+1}|\mathcal{F}_n] \leqslant (1 + a_n)T_n + \mathcal{A}_n - \mathcal{B}_n \qquad a.s.$$

where $a_n = 2Lm_4\gamma_n^2$, $\mathcal{A}_n = 2m_4\theta^*\gamma_n^2$ and $\mathcal{B}_n = 2\gamma_n\langle X_n - x^*, \nabla f(X_n)\rangle$. The four sequences $(T_n)$, $(a_n)$, $(\mathcal{A}_n)$ and $(\mathcal{B}_n)$ are positive sequences of random variables adapted to $(\mathcal{F}_n)$. We clearly have from (2) that

$$\sum_{n=1}^{\infty} a_n < +\infty \qquad \text{and} \qquad \sum_{n=1}^{\infty} \mathcal{A}_n < +\infty.$$

Hence, we deduce from the Robbins-Siegmund Theorem (Robbins & Siegmund, 1971) given by Theorem D.1 that $(T_n)$ converges a.s. towards a finite random variable $T$ and the series

$$\sum_{n=1}^{\infty} \mathcal{B}_n < +\infty \qquad a.s. \tag{25}$$

It only remains to show that $T = 0$ almost surely. Assume by contradiction that $T > 0$. For some positive constants $a < b$, denote by $\Omega$ the annulus of $\mathbb{R}^d$,

$$\Omega = \{x \in \mathbb{R}^d, \quad 0 < a < \|x - x^*\|^2 < b\}.$$

Let $F$ be the function defined, for all $x \in \mathbb{R}^d$, by

$$F(x) = \langle x - x^*, \nabla f(x)\rangle.$$

We have from Assumption 2 that $F$ is a continuous function in $\Omega$ compact. It implies that there exists a positive constant $c$ such that $F(x) > c$ for all $x \in \Omega$. However, for $n$ large enough, $X_n \in \Omega$, which ensures that $\gamma_n\langle X_n - x^*, \nabla f(X_n)\rangle > c\gamma_n$. Therefore, it follows from (25) that

$$\sum_{n=1}^{\infty} \gamma_n < +\infty.$$

This is of course in contradiction with the left side of conditions (2). Consequently, we conclude that $T = 0$ almost surely, leading to

$$\lim_{n \to +\infty} X_n = x^* \qquad a.s.$$

By continuity of the function $f$, we also have (8), which completes the proof of Theorem 1. $\qquad\square$

## 4.2 Central Limit Theorem

We carry on with the central limit theorem for the SCORS algorithm. In this subsection, we assume that $f$ is twice differentiable. Then, we denote by $H = \nabla^2 f(x^*)$ the Hessian matrix of $f$ at $x^*$.

**Assumption 5.** *Suppose that $f$ is twice differentiable with a unique equilibrium point $x^*$ in $\mathbb{R}^d$ such that $\nabla f(x^*) = 0$. Denote by $\rho = \lambda_{min}(H)$ the minimum eigenvalue of $H$. We assume that $\rho > 1/2$.*

The central limit theorem for the SCORS algorithm is as follows.

**Theorem 2.** *Consider that $(X_n)$ is the sequence generated by the SCORS algorithm with decreasing step $\gamma_n = 1/n$ and i.i.d. random direction vectors $(V_n)$ sharing the same distribution $\mathcal{P}$ satisfying Assumption 1. Moreover, suppose that Assumption 5 is satisfied and*

$$\lim_{n \to +\infty} X_n = x^* \qquad a.s. \tag{26}$$

*Then, we have the asymptotic normality*

$$\sqrt{n}(X_n - x^*) \quad \xrightarrow[n \to +\infty]{\mathcal{L}} \quad \mathcal{N}_d(0, \Sigma) \tag{27}$$

*where the asymptotic covariance matrix is given by*

$$\Sigma = \int_0^\infty (e^{-(H - \mathbf{I}_d/2)u})^T \Gamma e^{-(H - \mathbf{I}_d/2)u} du,$$

*with*

$$\Gamma = \mathbb{E}[VV^T Q V V^T], \tag{28}$$

*V is a random vector with the same distribution $\mathcal{P}$ and*

$$Q = \frac{1}{N} \sum_{k=1}^N \nabla f_k(x^*) \left(\nabla f_k(x^*)\right)^T. \tag{29}$$

*Proof.* The proof of Theorem 2 can be found in Appendix A. $\qquad\square$

We provide below the expressions of $\Gamma$ according to the direction distribution choice $\mathcal{P}$. The following result can be found in (Chen et al., 2024, p. 6).

**Proposition 3.** *Considering the matrix $Q$ defined in* (29) *and under the direction distribution $\mathcal{P}$ listed in Section 3, we have the following results,*

- **(U)** *Uniform in the canonical basis: $\Gamma^{(\mathbf{U})} = d \left(diag(Q)\right)$.*

- **(NU)** *Non-uniform in the canonical basis: $\Gamma^{(\mathbf{NU})} = diag\left(Q_{11}/p_1, \ldots, Q_{dd}/p_d\right)$.*

- **(G)** *Gaussian: $\Gamma^{(\mathbf{G})} = (2Q + tr(Q)\mathbf{I}_d)$.*

- **(S)** *Spherical: $\Gamma^{(\mathbf{S})} = \dfrac{d}{d+2}(2Q + tr(Q)\mathbf{I}_d)$.*

The central limit theorem is very useful and provides the theoretical guarantees necessary for building asymptotic confidence intervals based on the normal distribution. This result therefore makes it possible to address questions related to statistical inference such as the hypothesis testing while controlling the variability of algorithms. The point that seems difficult to us here lies on the computation of the asymptotic covariance matrix $\Sigma$ in a high-dimensional context. However, the standard Monte Carlo methods can be useful to obtain an estimation.

Furthermore, from Proposition 3, we can draw other interesting conclusions. In fact, one observes that the spherical random directions are always more efficient than the gaussian one. However, Chen et al. (2024) remark that there is no general domination relationship among the other choices. Nevertheless, in the particular case where $Q = \mathbf{I}_d$, we have $\Gamma^{(\mathbf{S})} = \Gamma^{(\mathbf{U})} = d\,\mathbf{I}_d$ which means that the spherical and the uniform choices achieve the same asymptotic performances for the central limit theorem. Thus, one can obtain different optimal choices of $\mathcal{P}$ according to the experimental design and the data used.

### 4.3  Non-asymptotic $\mathbb{L}^p$ rates of convergence

In this subsection, we are interesting by the non-asymptotic $\mathbb{L}^p$ rates of convergence of the SCORS algorithm. Hence, our goal is to investigate, for all integer $p \geqslant 1$, the convergence rate of $\mathbb{E}[\|X_n - x^*\|^{2p}]$ for the SCORS algorithm where the decreasing step is defined, for all $n \geqslant 1$ by,

$$\gamma_n = \frac{c}{n^\alpha}, \tag{30}$$

for some positive constant $c$ and $1/2 < \alpha \leqslant 1$.

**Assumption 6.** *Assume there exists a positive constant $\mu$ such that for all $x \in \mathbb{R}^d$ with $x \neq x^*$,*

$$\langle x - x^*, \nabla f(x) \rangle \geqslant \mu \|x - x^*\|^2.$$

This assumption is sometimes known as the Restricted Secant Inequality (RSI) in the literature (Karimi et al., 2016; Yi et al., 2020; Guille-Escuret et al., 2022). For instance, Theorem 2 in Karimi et al. (2016), established that when $f$ is a function with Lipschitz-continuous gradient, the RSI implies the Polyak-Lojasiewicz (PL) condition. However, the PL condition is not sufficient in our analysis to obtain the bound of $\mathbb{E}\left[\|X_n - x^*\|^2\right]$. In this case, we propose to use the RSI as minimal condition. We highlight that unlike the strong convexity, Assumption 6 alone does not even imply the convexity of $f$. The RSI is also weaker than the Weak Strong Convexity (Karimi et al., 2016; Yi et al., 2020).

First of all, we focus on the case $p = 1$ in Theorem 4 before extending this result for any integer $p \geqslant 1$ in Theorem 5.

**Theorem 4.** *Consider that $(X_n)$ is the sequence generated by the SCORS algorithm with decreasing step sequence $(\gamma_n)$ defined by (30). Suppose that Assumptions 1, 2, 4 and 6 are satisfied with $c\mu \leqslant 2^{\alpha-1}$ and $2c\mu > 1$ if $\alpha = 1$. Then, there exists a positive constant $K$ such that for all $n \geqslant 1$,*

$$\mathbb{E}\left[\|X_n - x^*\|^2\right] \leqslant \frac{K}{n^\alpha}. \tag{31}$$

*Proof.* The proof of Theorem 4 can be found in Appendix B. □

**Assumption 7.** *Assume that for some integer $p \geqslant 1$, there exists a positive constant $L_p$ such that for all $x \in \mathbb{R}^d$,*

$$\frac{1}{N} \sum_{k=1}^N \|\nabla f_k(x) - \nabla f_k(x^*)\|^{2p} \leqslant L_p \|x - x^*\|^{2p}.$$

*Moreover, suppose that there exists a positive constant $m_{4p}$ such that for all $n \geqslant 1$,*

$$\mathbb{E}[\|V_{n+1}\|^{4p}|\mathcal{F}_n] \leqslant m_{4p}. \tag{32}$$

**Theorem 5.** *Consider that $(X_n)$ is the sequence generated by the SCORS algorithm with decreasing step sequence $(\gamma_n)$ defined by (30) and such that the initial state $X_1$ belongs to $\mathbb{L}^{2p}$. Suppose that Assumptions 1, 2, 6 and 7 are satisfied with $pc\mu \leqslant 2^\alpha$ and $c\mu > 1$ if $\alpha = 1$. Then, there exists a positive constant $K_p$ such that for all $n \geqslant 1$,*

$$\mathbb{E}\left[\|X_n - x^*\|^{2p}\right] \leqslant \frac{K_p}{n^{p\alpha}}. \tag{33}$$

The proof of Theorem 5 is left to the reader as it follows the same lines as the proof of Theorem 4 in Bercu et al. (2024).

## 5 Numerical Experiments

The purpose of this section is to show the behavior of the SCORS algorithm on simulated data. For that goal, we consider the logistic regression model (Bach, 2014; Bercu et al., 2020) associated with the classical minimization problem $(\mathcal{P})$ of the convex function $f$ given, for all $x \in \mathbb{R}^d$, by

$$f(x) = \frac{1}{N} \sum_{k=1}^N f_k(x) = \frac{1}{N} \sum_{k=1}^N \left(\log(1 + \exp(\langle x, w_k \rangle)) - y_k \langle x, w_k \rangle\right),$$

where $x \in \mathbb{R}^d$ is a vector of unknown parameters, $w_k \in \mathbb{R}^d$ is a vector of features and the binary output $y_k \in \{0, 1\}$. One can easily see that the gradient of $f$ is given by

$$\nabla f(x) = \frac{1}{N} \sum_{k=1}^N \nabla f_k(x) = \frac{1}{N} \sum_{k=1}^N \left(\frac{1}{1 + \exp(-\langle x, w_k \rangle)} - y_k\right) w_k. \tag{34}$$

In the same way as Chen et al. (2024), we carry out the experiments on simulated data in order to illustrate the almost sure convergence (Theorem 1), the central limit theorem (Theorem 2) and the non-asymptotic $\mathbb{L}^2$ rate of convergence (Theorem 4). Therefore, we consider an independent and identically distributed collection $\{(w_1, y_1), \ldots, (w_N, y_N)\}$ where the covariate $w \sim \mathcal{N}_d(0, \mathbf{I}_d)$ and the response $y \in \{0, 1\}$ is sampled such that

$$\mathbb{P}(y = 1 | w) = \frac{1}{1 + e^{-\langle w, x^* \rangle}}. \tag{35}$$

The true model parameter $x^* \in \mathbb{R}^d$ is selected uniformly from the unit sphere. Furthermore, we set the sample size $N = 50000$ and the parameter dimension $d = 50$. The stepsize $\gamma_n = 1/n$ is used where $n \geqslant 1$ stands for the iterations. Here, we will compare the four methods $(\mathbf{U})$, $(\mathbf{NU})$, $(\mathbf{G})$ and $(\mathbf{S})$ described in Section 3. Let us define the initial value $g_{1,k}$ given, for any $k = 1, \ldots, N$, by $g_{1,k} = \nabla f_k(X_1)$. Moreover, the sequence $(g_{n,k})$ is updated, for all $n \geqslant 1$ and $1 \leqslant k \leqslant N$, as

$$g_{n+1,k} = \begin{cases} \nabla f_k(X_n) & \text{if } U_{n+1} = k, \\ g_{n,k} & \text{otherwise.} \end{cases} \tag{36}$$

For the non-uniform search distribution $(\mathbf{NU})$, the probabilities $p_{n,j}$ are computed for all $n \geqslant 2$ and $j \in [\![1, d]\!]$, as follows

$$p_{n,j} = \begin{cases} \dfrac{|g_1^{(j^*)}|}{\sum_{i=1}^{d} |g_1^{(i)}|} & \text{if } j = j^*, \\ \dfrac{1 - p_{n,j^*}}{d - 1} & \text{otherwise,} \end{cases} \tag{37}$$

where $j^* = \underset{j=1,\ldots,d}{\operatorname{argmax}} |g_1^{(j)}|$ and $g_1 = \sum_{k=1}^{N} g_{1,k}$.

The idea behind this choice is to select with the highest probability, the coordinate of a recursive estimate of the gradient which has the highest norm, since the goal is to reduce the gradient at each iteration in order to make it converge towards 0.

Figure 1 illustrates the almost sure convergence of the algorithms. This graph represents the relative optimality gap function $t \mapsto \|X_t - x^*\| \times \|X_1 - x^*\|^{-1}$, where $t$ stands for the number of the gradient coordinates computed. This criterion is chosen in order to make an honest comparison of the algorithms in competition. For instance, the SGD algorithm computes all $d$ coordinates of the gradient at each iteration, while the SCORS algorithm with uniform search distribution $(\mathbf{U})$ uses just a single coordinate. Thus, we made this choice to better take this principle into account.

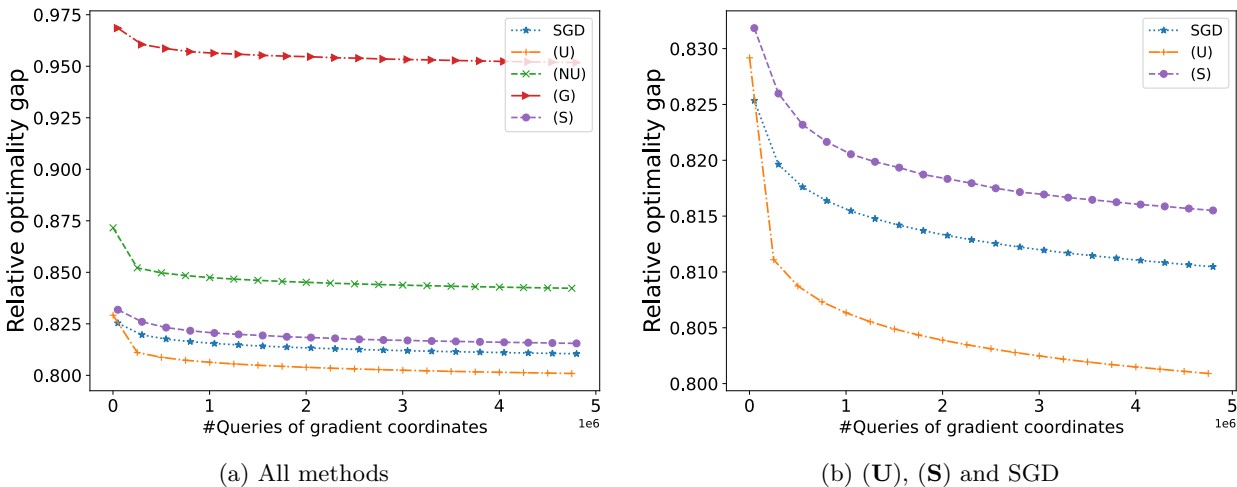

(a) All methods        (b) $(\mathbf{U})$, $(\mathbf{S})$ and SGD

Figure 1: Almost sure convergence of the algorithms with $\gamma_n = 1/n$.

As shown from the previous graph, the SCORS algorithm with uniform search distribution (**U**) is the best for this decreasing step. It is therefore interesting to see that the SCORS algorithm can achieve better performance than the SGD algorithm in terms of almost sure convergence. Moreover, we observe that the spherical distribution choice and the SGD algorithm are close. However, the gaussian choice is clearly the worst among all the methods in competition. Furthermore, the results concerning the central limit theorem of the (**U**), (**NU**), (**G**) and (**S**) algorithms are illustrated by Figure 2. We consider the same distributional convergence as Bercu et al. (2024) and use the standard Monte Carlo method to estimate the asymptotic standard deviation of each algorithm. We therefore observe that these simulations confirm the asymptotic normal distribution of the SCORS algorithms with all the search distribution choices considered. Note also that the (**U**) method has the smallest asymptotic standard deviation for the large dimension $d = 50$.

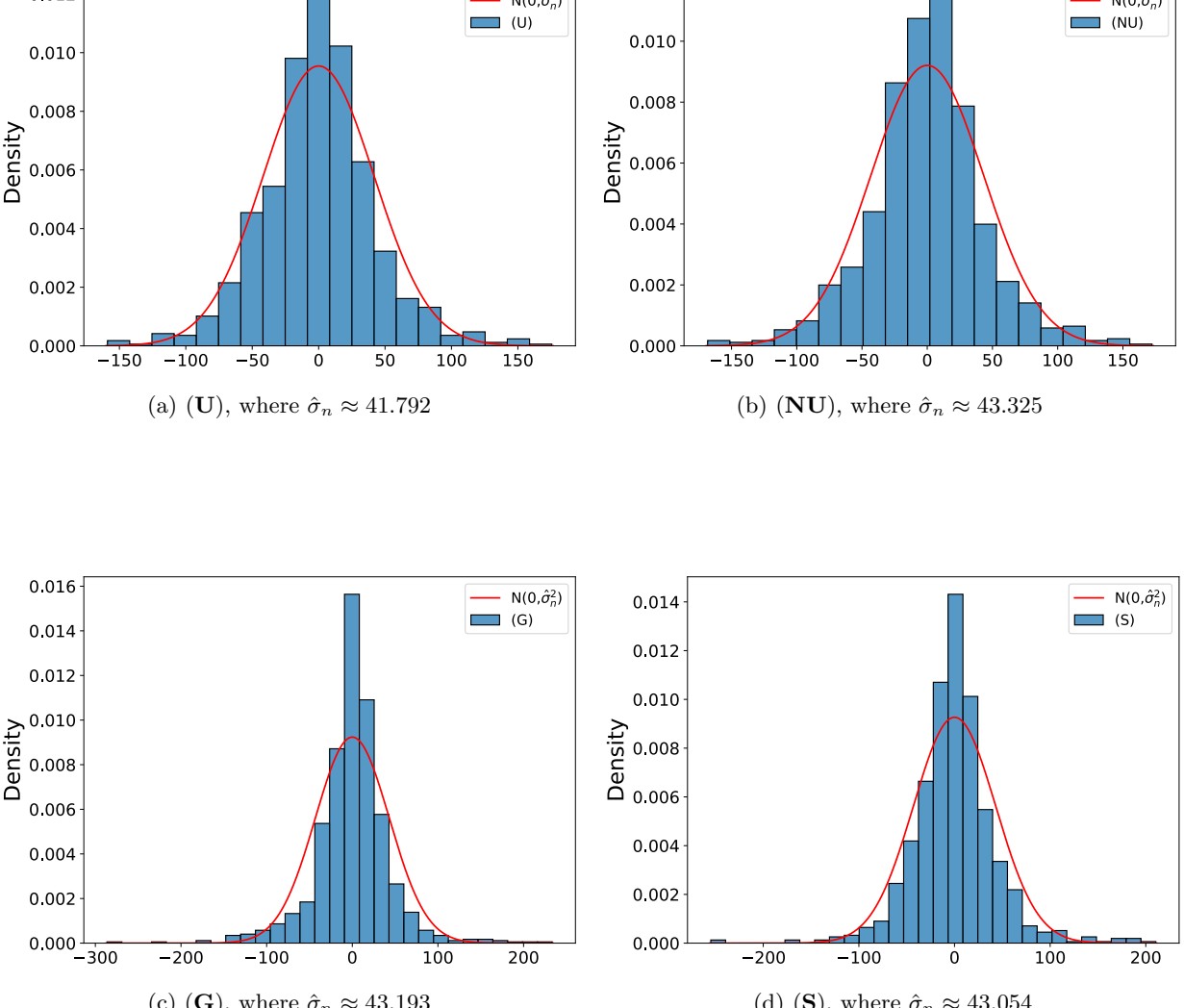

(a) (**U**), where $\hat{\sigma}_n \approx 41.792$

(b) (**NU**), where $\hat{\sigma}_n \approx 43.325$

(c) (**G**), where $\hat{\sigma}_n \approx 43.193$

(d) (**S**), where $\hat{\sigma}_n \approx 43.054$

Figure 2: We used 1000 samples, where each one was obtained by running the associated algorithm for $n = 500000$ iterations.

In Table 1, we report a comparison of the algorithm performances based on the computational cost. This criterion is estimated here by the average CPU time per iteration after running 5000000 iterations.

Table 1: CPU time per iteration (s).

| Algorithms | SGD | (**U**) | (**NU**) | (**G**) | (**S**) |
|---|---|---|---|---|---|
| CPU time | $5.05 \times 10^{-6}$ | $4.98 \times 10^{-6}$ | $12.01 \times 10^{-6}$ | $6.48 \times 10^{-6}$ | $8.92 \times 10^{-6}$ |

As expected, the SCORS algorithm with uniform search distribution (**U**) is the fastest in terms of computational time per iteration. Next come respectively the SGD, (**G**), (**S**) and (**NU**) methods. Nevertheless, the difference between the CPU times per iteration of the SGD and (**U**) algorithms is not very significant here, which is explained by the particular form of the gradient in this specific case of the logistic regression (34).

Lastly, Figure 3 provides approximate results on the non-asymptotic $\mathbb{L}^2$ rate of convergence. This graph just gives an idea on how the mean squared error of the algorithms decreases when $n$ goes to infinity. Each epoch consists of running 1000 iterations.

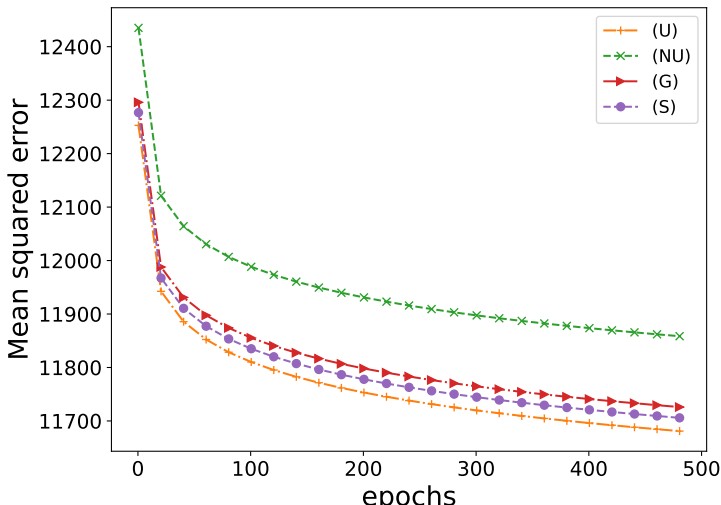

Figure 3: Mean squared error with respect to epochs. We confirm the decreasing order of the mean squared error of $X_n - x^*$ with respect to $n$.

### Acknowledgments

We would like to thank François Portier for fruitful discussions on preliminary version of the manuscript. This project has benefited from state support managed by the Agence Nationale de la Recherche (French National Research Agency) under the reference ANR-20-SFRI-0001.

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

## A   Proof of Theorem 2

*Proof.* Let us reformulate the SCORS iterates for all $n \geqslant 1$, as follows

$$X_{n+1} = X_n - \frac{1}{n}\left(\nabla f(X_n) + \varepsilon_{n+1}\right), \tag{38}$$

with

$$\varepsilon_{n+1} = \mathcal{Y}_{n+1} - \nabla f(X_n), \tag{39}$$

and

$$\mathcal{Y}_{n+1} = D(V_{n+1})\nabla f_{U_{n+1}}(X_n). \tag{40}$$

Moreover, we recall from (13) in the proof of Theorem 1 that

$$\mathbb{E}[\mathcal{Y}_{n+1}|\mathcal{F}_n] = \nabla f(X_n) \qquad a.s. \tag{41}$$

Then, it follows from (39) and (41) that

$$\mathbb{E}[\varepsilon_{n+1}|\mathcal{F}_n] = 0 \qquad a.s., \tag{42}$$

which means that $(\varepsilon_n)$ is a martingale difference sequence adapted to the filtration $(\mathcal{F}_n)$. Therefore, we can also deduce that almost surely

$$\mathbb{E}[\varepsilon_{n+1}\varepsilon_{n+1}^T|\mathcal{F}_n] = \mathbb{E}[\mathcal{Y}_{n+1}\mathcal{Y}_{n+1}^T|\mathcal{F}_n] - \nabla f(X_n)\left(\nabla f(X_n)\right)^T. \tag{43}$$

Since $X_n$ converges towards $x^*$ almost surely, we obtain from Assumption 5 that

$$\lim_{n\to+\infty} \nabla f(X_n) = 0 \qquad a.s., \tag{44}$$

which implies that

$$\lim_{n\to+\infty} \mathbb{E}[\varepsilon_{n+1}\varepsilon_{n+1}^T|\mathcal{F}_n] = \lim_{n\to+\infty} \mathbb{E}[\mathcal{Y}_{n+1}\mathcal{Y}_{n+1}^T|\mathcal{F}_n] \qquad a.s. \tag{45}$$

However, we recall by definition (40) that

$$\begin{aligned}
\mathbb{E}[\mathcal{Y}_{n+1}\mathcal{Y}_{n+1}^T|\mathcal{F}_n] &= \mathbb{E}[D(V_{n+1})\nabla f_{U_{n+1}}(X_n)\left(D(V_{n+1})\nabla f_{U_{n+1}}(X_n)\right)^T|\mathcal{F}_n]\\
&= \mathbb{E}[V_{n+1}V_{n+1}^T\nabla f_{U_{n+1}}(X_n)\left(\nabla f_{U_{n+1}}(X_n)\right)^T V_{n+1}V_{n+1}^T|\mathcal{F}_n].
\end{aligned} \tag{46}$$

Hereafter, we define $\mathcal{G}_{n+1} = \sigma(X_1,\ldots,X_n,V_{n+1})$. It is obvious that $\mathcal{F}_n \subset \mathcal{G}_{n+1}$. Therefore, by the tower property of the conditional expectation and since $V_{n+1}$ is $\mathcal{G}_{n+1}$-measurable, we obtain from (46) that

$$\mathbb{E}[\mathcal{Y}_{n+1}\mathcal{Y}_{n+1}^T|\mathcal{F}_n] = \mathbb{E}\left[V_{n+1}V_{n+1}^T\mathbb{E}\left[\nabla f_{U_{n+1}}(X_n)\left(\nabla f_{U_{n+1}}(X_n)\right)^T|\mathcal{G}_{n+1}\right]V_{n+1}V_{n+1}^T\Big|\mathcal{F}_n\right]. \tag{47}$$

Furthermore, we have that

$$\mathbb{E}\left[\nabla f_{U_{n+1}}(X_n)\left(\nabla f_{U_{n+1}}(X_n)\right)^T|\mathcal{G}_{n+1}\right] = \frac{1}{N}\sum_{k=1}^{N}\nabla f_k(X_n)\left(\nabla f_k(X_n)\right)^T. \tag{48}$$

Thus, it follows from (47) and (48) that

$$\mathbb{E}[\mathcal{Y}_{n+1}\mathcal{Y}_{n+1}^T|\mathcal{F}_n] = \mathbb{E}\left[V_{n+1}V_{n+1}^T Q_n V_{n+1}V_{n+1}^T|\mathcal{F}_n\right]. \tag{49}$$

where

$$Q_n = \frac{1}{N}\sum_{k=1}^{N}\nabla f_k(X_n)\left(\nabla f_k(X_n)\right)^T. \tag{50}$$

Let us consider the symmetric matrix $Q$ defined by

$$Q = \frac{1}{N} \sum_{k=1}^{N} \nabla f_k(x^*) \left(\nabla f_k(x^*)\right)^T. \tag{51}$$

In fact, we have from the Jensen inequality that

$$
\begin{aligned}
&\left\| \mathbb{E}\left[ V_{n+1} V_{n+1}^T Q_n V_{n+1} V_{n+1}^T | \mathcal{F}_n \right] - \mathbb{E}\left[ V_{n+1} V_{n+1}^T Q V_{n+1} V_{n+1}^T \right] \right\| \\
&= \left\| \mathbb{E}\left[ V_{n+1} V_{n+1}^T Q_n V_{n+1} V_{n+1}^T | \mathcal{F}_n \right] - \mathbb{E}\left[ V_{n+1} V_{n+1}^T Q V_{n+1} V_{n+1}^T | \mathcal{F}_n \right] \right\| \\
&= \left\| \mathbb{E}\left[ V_{n+1} V_{n+1}^T (Q_n - Q) V_{n+1} V_{n+1}^T | \mathcal{F}_n \right] \right\| \\
&\leqslant \mathbb{E}\left[ \| V_{n+1} V_{n+1}^T (Q_n - Q) V_{n+1} V_{n+1}^T \| | \mathcal{F}_n \right] \\
&\leqslant \mathbb{E}\left[ \| V_{n+1} \|^4 \| Q_n - Q \| | \mathcal{F}_n \right]. 
\end{aligned} \tag{52}
$$

However, $Q_n$ is $\mathcal{F}_n$-measurable and $V_{n+1}$ is independent from $\mathcal{F}_n$. Therefore, Assumption 1 with the inequality (52), imply that there exists a positive constant $m_4$ such that almost surely

$$\left\| \mathbb{E}\left[ V_{n+1} V_{n+1}^T Q_n V_{n+1} V_{n+1}^T | \mathcal{F}_n \right] - \mathbb{E}\left[ V_{n+1} V_{n+1}^T Q V_{n+1} V_{n+1}^T \right] \right\| \leqslant m_4 \| Q_n - Q \|. \tag{53}$$

Once again from the almost sure convergence of $X_n$ towards $x^*$ and Assumption 5, we obtain that

$$\lim_{n \to +\infty} \| Q_n - Q \| = 0 \qquad a.s. \tag{54}$$

Hence, we deduce from (53) and (54) that

$$\lim_{n \to +\infty} \left\| \mathbb{E}\left[ V_{n+1} V_{n+1}^T Q_n V_{n+1} V_{n+1}^T | \mathcal{F}_n \right] - \mathbb{E}\left[ V_{n+1} V_{n+1}^T Q V_{n+1} V_{n+1}^T \right] \right\| = 0 \qquad a.s. \tag{55}$$

which leads to

$$\lim_{n \to +\infty} \mathbb{E}\left[ V_{n+1} V_{n+1}^T Q_n V_{n+1} V_{n+1}^T | \mathcal{F}_n \right] = \Gamma \qquad a.s., \tag{56}$$

where

$$\Gamma = \mathbb{E}\left[ V V^T Q V V^T \right]. \tag{57}$$

Thus, combining the three contributions (45), (49) and (56), it follows that

$$\lim_{n \to +\infty} \mathbb{E}[\varepsilon_{n+1} \varepsilon_{n+1}^T | \mathcal{F}_n] = \Gamma \qquad a.s. \tag{58}$$

Therefore, we deduce from Toeplitz's lemma that

$$\lim_{n \to +\infty} \frac{1}{n} \sum_{k=1}^{n} \mathbb{E}[\varepsilon_k \varepsilon_k^T | \mathcal{F}_{k-1}] = \Gamma \qquad a.s. \tag{59}$$

Furthermore, we have for all $n \geqslant 1$

$$\| \varepsilon_{n+1} \|^2 \leqslant 2 \left( 1 + \| V_{n+1} \|^4 \right) C_n, \tag{60}$$

where

$$C_n = \max_{j=1,\dots,N} \| \nabla f_j(X_n) \|^2. \tag{61}$$

Therefore, we define for all $\epsilon > 0$ and some positive constant M,

$$A_n = \frac{1}{n} \sum_{k=1}^{n} \mathbb{E}\left[ \| \varepsilon_k \|^2 \mathbb{1}_{\{\| \varepsilon_k \| \geqslant \epsilon \sqrt{n}\}} \mathbb{1}_{\{\| V_k \| \leqslant M\}} | \mathcal{F}_{k-1} \right], \tag{62}$$

and

$$B_n = \frac{1}{n} \sum_{k=1}^{n} \mathbb{E}\left[\|\varepsilon_k\|^2 \mathbb{1}_{\{\|\varepsilon_k\| \geqslant \epsilon\sqrt{n}\}} \mathbb{1}_{\{\|V_k\| > M\}} | \mathcal{F}_{k-1}\right]. \tag{63}$$

We obtain from (60) that

$$\begin{aligned}
A_n &= \frac{1}{n} \sum_{k=1}^{n} \mathbb{E}\left[\frac{\|\varepsilon_k\|^4}{\|\varepsilon_k\|^2} \mathbb{1}_{\{\|\varepsilon_k\| \geqslant \epsilon\sqrt{n}\}} \mathbb{1}_{\{\|V_k\| \leqslant M\}} | \mathcal{F}_{k-1}\right] \\
&\leqslant \frac{1}{\epsilon^2 n^2} \sum_{k=1}^{n} \mathbb{E}\left[\|\varepsilon_k\|^4 \mathbb{1}_{\{\|V_k\| \leqslant M\}} | \mathcal{F}_{k-1}\right] \\
&\leqslant \frac{4(1+M^4)^2}{\epsilon^2 n^2} \sum_{k=1}^{n} \mathbb{E}\left[C_{k-1}^2 | \mathcal{F}_{k-1}\right] \\
&\leqslant \frac{4(1+M^4)^2}{\epsilon^2 n} \sup_{n \geqslant 1}\left\{C_n^2\right\}.
\end{aligned} \tag{64}$$

Since $X_n$ converges towards $x^*$, it follows with Assumption 5 that

$$\lim_{n \to +\infty} C_n = \max_{j=1,\dots,N} \|\nabla f_j(x^*)\|^2 < +\infty \qquad a.s. \tag{65}$$

Then, we have that almost surely

$$\sup_{n \geqslant 1}\left\{C_n^2\right\} < +\infty, \tag{66}$$

which immediately implies with (64) that

$$\lim_{n \to +\infty} A_n = 0 \qquad a.s. \tag{67}$$

Moreover, we obtain with the same inequality (60) that almost surely

$$\begin{aligned}
B_n &\leqslant \frac{1}{n} \sum_{k=1}^{n} \mathbb{E}\left[\|\varepsilon_k\|^2 \mathbb{1}_{\{\|V_k\| > M\}} | \mathcal{F}_{k-1}\right] \\
&\leqslant \frac{2}{n} \sum_{k=1}^{n} \mathbb{E}\left[\left(1 + \|V_k\|^4\right) C_{k-1} \mathbb{1}_{\{\|V_k\| > M\}} | \mathcal{F}_{k-1}\right] \\
&\leqslant \frac{2}{n} \sum_{k=1}^{n} \mathbb{E}\left[\left(1 + \|V_k\|^4\right) \mathbb{1}_{\{\|V_k\| > M\}} | \mathcal{F}_{k-1}\right] C_{k-1} \\
&\leqslant \left(\frac{2}{n} \sum_{k=1}^{n} C_{k-1}\right) \mathbb{E}\left[\left(1 + \|V\|^4\right) \mathbb{1}_{\{\|V\| > M\}}\right].
\end{aligned} \tag{68}$$

By using Toeplitz's lemma and the contribution (65), we deduce that

$$\lim_{n \to +\infty} \left(\frac{1}{n} \sum_{k=1}^{n} C_{k-1}\right) = \max_{j=1,\dots,N} \|\nabla f_j(x^*)\|^2 < +\infty \qquad a.s. \tag{69}$$

It follows from (68) and (69) that for any positive constant M,

$$\limsup_{n \to +\infty} B_n \leqslant 2 \left(\max_{j=1,\dots,N} \|\nabla f_j(x^*)\|^2\right) \mathbb{E}\left[\left(1 + \|V\|^4\right) \mathbb{1}_{\{\|V\| > M\}}\right] \qquad a.s. \tag{70}$$

Nevertheless, we obtain from the Lebesgue dominated convergence theorem that

$$\lim_{M \to +\infty} \mathbb{E}\left[\left(1 + \|V\|^4\right) \mathbb{1}_{\{\|V\| > M\}}\right] = 0. \tag{71}$$

Hence, it implies with (70) that

$$\lim_{n \to +\infty} B_n = 0 \qquad a.s. \tag{72}$$

Consequently, by putting together the contributions (67) and (72), we immediately deduce that for all $\epsilon > 0$,

$$\lim_{n \to +\infty} \frac{1}{n} \sum_{k=1}^{n} \mathbb{E}\left[ \|\varepsilon_k\|^2 \mathbb{1}_{\{\|\varepsilon_k\| \geqslant \epsilon\sqrt{n}\}} | \mathcal{F}_{k-1} \right] = 0 \qquad a.s.$$

Finally, it follows from the central limit theorem for stochastic algorithms given by Theorem 2.3 in (Zhang, 2016) that

$$\sqrt{n}(X_n - x^*) \xrightarrow[n \to +\infty]{\mathcal{L}} \mathcal{N}_d(0, \Sigma),$$

where

$$\Sigma = \int_0^\infty (e^{-(H - \mathbf{I}_d/2)u})^T \Gamma e^{-(H - \mathbf{I}_d/2)u} du,$$

which completes the proof of Theorem 2. $\qquad\square$

## B   Proof of Theorem 4

*Proof.* We already proved in (23) that for all $n \geq 1$,

$$\mathbb{E}[T_{n+1}|\mathcal{F}_n] \leqslant T_n - 2\gamma_n\langle X_n - x^*, \nabla f(X_n)\rangle + 2m_4\gamma_n^2(\tau^2(X_n) + \theta^*) \qquad a.s. \tag{73}$$

Therefore, it follows from Assumption 6 that $\langle X_n - x^*, \nabla f(X_n)\rangle \geqslant \mu T_n$, which leads to

$$\mathbb{E}[T_{n+1}|\mathcal{F}_n] \leqslant (1 - 2\mu\gamma_n)T_n + 2m_4\gamma_n^2(\tau^2(X_n) + \theta^*) \qquad a.s. \tag{74}$$

By taking the expectation on both side of this inequality, we obtain that for all $n \geq 1$,

$$\mathbb{E}[T_{n+1}] \leqslant (1 - 2\mu\gamma_n)\mathbb{E}[T_n] + 2m_4\gamma_n^2\left(\mathbb{E}[\tau^2(X_n)] + \theta^*\right). \tag{75}$$

Hence, we deduce from Corollary 7 in Appendix C below that there exists a positive constant $b_1$ such that, for all $n \geqslant 1$, $\mathbb{E}[\tau^2(X_n)] \leqslant b_1$. Consequently, the inequality (75) immediately implies, for all $n \geqslant 1$, that

$$\mathbb{E}[T_{n+1}] \leqslant \left(1 - \frac{a}{(n+1)^\alpha}\right)\mathbb{E}[T_n] + \frac{b}{(n+1)^{2\alpha}} \tag{76}$$

where $a = 2\mu c$ and $b = c^2 2^{2\alpha+1} m_4(b_1 + \theta^*)$. Finally, we can conclude from Lemma D.1 that there exists a positive constant $K$ such that for any $n \geqslant 1$,

$$\mathbb{E}[\|X_n - x^*\|^2] \leqslant \frac{K}{n^\alpha},$$

which achieves the proof of Theorem 4. $\qquad\square$

## C   Additional asymptotic result on the convergence in $\mathbb{L}^{2p}$

The purpose of this appendix is to provide additional asymptotic properties of the SCORS algorithm that will be useful in the proofs of our main results. First of all, we recall for some integer $p \geqslant 1$ that $T_n^p = \|X_n - x^*\|^{2p}$.

**Theorem 6.** *Consider that $(X_n)$ is the sequence generated by the SCORS algorithm with decreasing step $\gamma_n$ satisfying (2). Suppose that Assumptions 1, 2, 6 and 7 are satisfied. Then, we have that*

$$\sum_{n=1}^\infty \gamma_n T_n^p < +\infty \qquad a.s., \tag{77}$$

*and*

$$\sum_{n=1}^\infty \gamma_n \mathbb{E}[T_n^p] < +\infty. \tag{78}$$

The proof of Theorem 6 is very analogous to that of Theorem 8 in Bercu et al. (2024). A direct consequence of Theorem 6, using the left-hand side of (2), is as follows.

**Corollary 7.** *Assume that the conditions of Theorem 6 hold. Then, for all $p \geqslant 1$, we have*

$$\lim_{n \to +\infty} T_n^p = 0 \qquad a.s.,$$

*and*

$$\lim_{n \to +\infty} \mathbb{E}[T_n^p] = 0.$$

## D    Some useful existing results

We first recall the well-known Robbins-Siegmund Theorem (Robbins & Siegmund, 1971).

**Theorem D.1** (Robbins-Siegmund theorem)**.** *Let $(T_n), (a_n), (\mathcal{A}_n), (\mathcal{B}_n)$ be four positive sequences of random variables adapted to a filtration $(\mathcal{F}_n)$ such that*

$$\mathbb{E}[T_{n+1}|\mathcal{F}_n] \leqslant (1 + a_n)T_n + \mathcal{A}_n - \mathcal{B}_n,$$

*where*

$$\sum_{n=1}^{\infty} a_n < +\infty \qquad and \qquad \sum_{n=1}^{\infty} \mathcal{A}_n < +\infty \qquad a.s.$$

*Then, $(T_n)$ converges almost surely towards a finite random variable $T$ and*

$$\sum_{n=1}^{\infty} \mathcal{B}_n < +\infty \qquad a.s.$$

The next lemma provide very useful inequality for non-asymptotic convergence rates. This result is given by Lemma A.3 in supplementary material of Bercu & Bigot (2021), see also Theorem 1 in Bach & Moulines (2011).

**Lemma D.1.** *(Bercu & Bigot, 2021). Let $(Z_n)$ be a sequence of positive real numbers satisfying, for all $n \geqslant 1$, the recursive inequality*

$$Z_{n+1} \leqslant \left(1 - \frac{a}{(n+1)^\alpha}\right) Z_n + \frac{b}{(n+1)^\beta}, \tag{79}$$

*where $a, b, \alpha$ and $\beta$ are positive constants satisfying $a \leqslant 2^\alpha$, $\alpha \leqslant 1$, $1 < \beta < 2$ and $\beta \leqslant 2\alpha$ with $\beta < a + 1$ in the special case where $\alpha = 1$. Then, there exists a positive constant $C$ such that, for any $n \geqslant 1$,*

$$Z_n \leqslant \frac{C}{n^{\beta - \alpha}}. \tag{80}$$

