# OpenReview forum: "A stochastic gradient descent algorithm with random search directions"
_TMLR — Accepted by TMLR_

### Review · Reviewer_NNts · 2025-05-11

**Summary Of Contributions:**

This paper extends stochastic coordinate gradient descent by allowing each iteration to select an arbitrary direction vector—rather than a single coordinate—and perform the update along that direction. Under mild regularity conditions and a suitably chosen step-size schedule, the authors prove almost-sure convergence of the method. They further establish both asymptotic convergence rates and non-asymptotic $\mathbb{L}^p$ convergence bounds under additional standard assumptions on the step sizes.

**Audience:**

Yes

**Broader Impact Concerns:**

I don't think there're any ethical concerns of this paper.

**Claims And Evidence:**

Yes

**Requested Changes:**

Here’re some changes I would like to see from the revision of the paper.
1. More related works and discussions on the theorems as I mentioned before in the "Weakness" section.
2. For the experiments confirming the central limit theorem, compute the theoretical variance and compare it with the empirical variance. Similarly, for the non-asymptotic experiments, ideally show a comparison between theoretical and empirical rates, but it may be hard to compute the theoretical rates here (and it's only an upper bound).

Here're some suggestions I think could make this paper better.
1. In the abstract and introduction, the sentence
   > “In this paper, we develop a new class of stochastic gradient descent algorithms with random search directions which uses the directional derivative of the gradient estimate following more general random vectors.”

   is too long and confusing. Consider rewriting it for clarity and brevity.
2. Use an enumeration in the introduction to emphasize your contributions; this will make them stand out more clearly.
3. Assumption 1 combines unbiasedness and boundedness; since these are used separately in the proofs, split them into two distinct assumptions. Also, clarify “the 4th conditional moment of $V_n$ is bounded” by providing the exact mathematical expression, noting that $V_n$ is vector-valued.
4. Specify that every vector norm is the $\ell_2$ norm, to avoid any ambiguity.
5. Instead of stating “the rest of the proof follows as in …”, either include the full proof or add intermediate lemmas to make the paper more self-contained.

Here're some questions I hope the authors could address in the revision of the paper:
1. In the proof of Theorem 1, you mentioned $T_n$ is a Lyapunov function, but where did you use this property. Is it omitted at the end of the proof?
2. In Assumption 5, why is the minimum eigenvalue should be greater than $\frac{1}{2}$? This $\frac{1}{2}$ seems to be quite arbitrary here and I couldn't find where it is used.
3. What's $V$ in (27)? It's actually not defined. Since each $V_n$ is i.i.d. you may want to simplify your notation.
4. Why is Assumption 3 needed for Theorem 1 but not for Theorem 4 and 5? This is a bit surprising to me.
5. In your experiments, why are $\\{(w_1,y_1),\cdots,(w_N,y_N)\\}$ chosen i.i.d. with Gaussian distribution? I don't think in your theorem there isn't much restrictions on $f_k$.
6. I don't understand your relative optimality gap function. Isn't the initial point $X_1$ the same for different methods? If so, how is this different from just using $\lVert X_t-x^*\rVert$? Why does this metric make the comparison fair?
7. All of your experiments shows uniform is the best, so what's the point of using other distributions?

**Strengths And Weaknesses:**

Strengths
1. The paper introduces novel and interesting results—for example, a central limit theorem for stochastic coordinate gradient descent—which are likely to be of broad interest to the optimization community.
2. The technical exposition is clear and well-organized. Even readers who are not specialists in this area will find the presentation straightforward and easy to follow.

Weakness:
1. I think there could be more discussion on related works, especially works in the past two years. For example, this paper (https://arxiv.org/abs/2307.01169) by Ramesh et. al. studied greedy block coordinate gradient descent and may be relevant.
2. The latter sections of the paper are overly terse: they state theorems and definitions without sufficient commentary on their implications or on the necessity and reasonableness of the underlying assumptions. A deeper discussion of how these results connect to or extend prior work would strengthen the manuscript.
3. The experimental section adds limited value. Since the theoretical results are straightforward and convincing, it is unclear what additional insight the current experiments provide—and they may not even adequately demonstrate the theorems’ claims.

---

> ### Author Response · Authors · 2025-06-13
>
> Thank you for your valuable feedback on our manuscript. In the following, we give detailed responses to each points.
>
> First of all, we have chosen to add more discussions on related works. Secondly, on the central limit theorem, we are not able to compute the theoretical covariance matrix $\Sigma$ in a high-dimensional setting.
> The reasons are among others because it is a matrix integral and its expression depends in a complex way on $x^{*}$, the minimizer of $f$
> which is unknown in principle. The covariance matrix $\Sigma$ is also the solution of a Lyapunov equation but it is not possible to produce an explicit solution of this Lyapunov equation.
> That is the reason why we propose to use the standard Monte Carlo methods in order to have an estimation of $\Sigma$. Concerning the non-asymptotic convergence rates, our results are intended to establish the linear convergence rate of the SCORS algorithms. As you mentioned, these theoretical rates are upper bounds and unfortunately hard to compute since the calculation of the constant $K$ is quite complicated.
>
> > 1. In the proof of Theorem 1, you mentioned $T_n$ is a Lyapunov function, but where did you use this property. Is it omitted at the end of the proof?
>
> The Lyapunov function $T_n$ allows us to obtain the inequality (24) which is necessary for the application of the Robbins-Siegmund theorem and the deduction of the almost sure convergence. We have now added the full end of the proof to better understand its importance.
>
> > 2. In Assumption 5, why is the minimum eigenvalue should be greater than 1/2 ? This 1/2 seems to be quite arbitrary here and I couldn't find where it is used.
>
> It is a standard condition in central limit theorem theory for stochastic algorithms. Therefore, this technical condition is usually assumed and appears necessary to obtain the normal distribution in the central limit theorem. Without this assumption, the algorithm will converge towards degenerate distributions. We used Theorem 2.3 from Zhang (2016, page 7) where this condition appears. We have chosen to add a remark in order to highlight the role played by this hypothesis.
>
> > 3. What's $V$ in (27)? It's actually not defined. Since each $V_n$ is i.i.d. you may want to simplify your notation.
>
> We just use $V$ to denote a random vector with the same distribution as $V_n$ because they are i.i.d. random vectors, and in order to simplify the notation. We have now added a comment to define $V$ for clarity.
>
> > 4. Why is Assumption 3 needed for Theorem 1 but not for Theorem 4 and 5? This is a bit surprising to me.
>
> In Theorems 4 and 5, we replaced Assumption 3 by Assumption 6, a slightly stronger version. We needed this new assumption to properly handle the dot product in the inequality (72). This technical condition is also necessary to deduce the non-asymptotic rates of convergence. Finally, we can also see that when Assumption 6 is satisfied, Assumption 3 will also be satisfied.
>
> > 5. In your experiments, why are {$(w_1,y_1),\dots,(w_N,y_N)$}  chosen i.i.d. with Gaussian distribution? I don't think in your theorem there isn't much restrictions on $f_k$.
>
> This was only an arbitrary choice. Our theorems do not need such restrictions on the functions $f_k$.
>
> > 6. I don't understand your relative optimality gap function. Isn't the initial point $X_1$ the same for different methods? If so, how is this different from just using $\lVert X_t-x^{*}\rVert$? Why does this metric make the comparison fair?
>
> We made this choice to take into account the gap of the initial point and visualize relative distance. However, since we used the same initial point $X_1$ for all methods, consider $\lVert X_t -x^{*} \rVert$, will absolutely not change the comparison and will keep the same conclusions.
>
> > 7. All of your experiments shows uniform is the best, so what's the point of using other distributions?
>
> Our experiments suggest that uniform is the best for this particular loss function with the data used.
> Nevertheless, from Proposition 3, we are not able to prove that it will be always the case because there is not general domination relationship between $\Gamma$ matrices. Therefore, we have maintained the other possible choices of random search direction, firstly to propose a more general theory of the stochastic coordinates algorithms and secondly to keep in mind that different optimal choices can be obtained in other contexts.

---

### Review · Reviewer_T6L8 · 2025-05-15

**Summary Of Contributions:**

#Summary

In this paper, the authors consider the problem of stochastic optimization and propose the stochastic coordinate gradient descent algorithm with random search direction (SCORS).  The main motivation of the proposed algorithm is to address the limitation of the existing stochastic coordinate descent (SCD) algorithm. More specifically, SCD actually could be viewed as an efficient variant of the classical stochastic gradient descent (SGD) algorithm in some cases. However, as the authors pointed out, the coordinated sampling policy of SCD is limited. Moreover, the authors have proved the almost sure convergence, the central limit theorem, and the non-asymptotic convergence for the proposed algorithm. Experimental results have also been provided to verify these theoretical guarantees.

**Audience:**

Yes

**Broader Impact Concerns:**

This is a theoretical paper, and I do not find any ethical issues.

**Claims And Evidence:**

Yes

**Requested Changes:**

Please see Strengths And Weaknesses

**Strengths And Weaknesses:**

#Major Concerns
1) Although the proposed algorithm seems more general than the existing SCD algorithm, from both theories and experiments, I cannot find benefits of this generalization. Specifically, the proposed algorithm provides different choices of the coordinated sampling policy, but the uniform sampling (which reduces to the naive SCD) achieves the best experimental results. Moreover, from Proposition 3, it is unclear whether other choices of sampling policy can achieve a better central limit theorem than uniform sampling.
2) Although the almost sure convergence, the central limit theorem, and the non-asymptotic convergence for the proposed algorithm have been proved, the value of these theoretical results for the (optimization and machine learning) community is unclear. It would be better if the authors could highlight their theoretical contributions.
3) Although the authors claim that "our Assumption 4 is of key importance and essential in general non-convex setting", some assumptions (e.g., Assumptions 3, 5, and 6) used in this paper actually make the analysis easier than the general non-convex case. Specifically, it seems that Assumption 5 implies the loss function is strongly convex.
4) In the experiments, it seems that the step size of all algorithms is set to be $\gamma_n=1/n$. However, for the convex loss function utilized in the experiments, the commonly utilized step size of SGD should be $O(1/\sqrt{n})$. It is unclear whether such a choice of step size affects the comparison of these algorithms, especially the mean squared error shown in Figure 3.

#Minor Concerns
1) I am not sure whether the frequent use of "almost surely" is necessary.
2) The authors have referred the readers to learn some proofs from Bercu et al. (2024). It would be better if a sketch of such proofs could be provided in this paper.
3) I prefer "the assumptions used in Leluc & Portier (2022)" to "the assumptions used in (Leluc & Portier, 2022)".

---

> ### Author Response · Authors · 2025-06-13
>
> Thank you very much for taking the time to review our article. In the following, we give detailed responses to each points.
>
> > 1. Although the proposed algorithm seems more general than the existing SCD algorithm, from both theories and experiments, I cannot find benefits of this generalization. Specifically, the proposed algorithm provides different choices of the coordinated sampling policy, but the uniform sampling (which reduces to the naive SCD) achieves the best experimental results. Moreover, from Proposition 3, it is unclear whether other choices of sampling policy can achieve a better central limit theorem than uniform sampling.
>
> You can observe that the definition of the asymptotic covariance matrix $\Sigma$ in Theorem 2, is exactly the same for all search directions, except for the matrix $\Gamma$ which depends on the choices of the direction distribution. Consequently, a "small" choice of $\Gamma$ yields to a better value of the asymptotic covariance $\Sigma$. Unfortunately, from Proposition 3, we are not able to prove that the uniform is always the best because there is not general domination relationship between $\Gamma$ matrices.
>
> > 2. Although the almost sure convergence, the central limit theorem, and the non-asymptotic convergence for the proposed algorithm have been proved, the value of these theoretical results for the (optimization and machine learning) community is unclear. It would be better if the authors could highlight their theoretical contributions.
>
> Our central limit theorem result, ensures that there exists a CLT with normal distribution and provides all the theoretical guarantees for using the statistical inferential tools such as the asymptotic confidence intervals and hypothesis testing on the estimated parameters.
>
> > 3. Although the authors claim that "our Assumption 4 is of key importance and essential in general non-convex setting", some assumptions (e.g., Assumptions 3, 5, and 6) used in this paper actually make the analysis easier than the general non-convex case. Specifically, it seems that Assumption 5 implies the loss function is strongly convex.
>
> You can observe that Assumption 5 does not imply that the loss function is strongly convex. In fact, this condition is on the minimum eigenvalue of $H=\nabla^2 f ( x^* )$, the Hessian matrix of $f$ only at point $x^*$. This condition is not required for all vectors in $\mathbb{R}^d$.
>
> > 4. In the experiments, it seems that the step size of all algorithms is set to be $\gamma_n=1/n$ . However, for the convex loss function utilized in the experiments, the commonly utilized step size of SGD should be $\mathcal{O}(1/\sqrt{n})$. It is unclear whether such a choice of step size affects the comparison of these algorithms, especially the mean squared error shown in Figure 3.
>
> From a theoretical point of view, the step size $\gamma_n=1/\sqrt{n}$ does not satisfy the standard conditions $\sum_{n=1}^\infty \gamma_n = + \infty$ and $\sum_{n=1}^\infty \gamma_n^2 < + \infty$ used to apply the Robbins-Siegmund theorem. Moreover, the non-asymptotic $\mathbb{L}^p$ rates of convergence given in Theorems 4 and 5, show that the step size $\mathcal{O}(1/n)$ achieves clearly the best rate. Finally, we used $\gamma_n=1/n$ in our experiments because our central limit theorem is valid only for this choice.

---

> > ### Comment · Reviewer_T6L8 · 2025-06-27
> >
> > Thank the authors for the explanation. However, I am still not convinced that the proposed algorithm has a significant advantage over the naive SCD in both theory and practice.

---

### Review · Reviewer_evch · 2025-06-07

**Summary Of Contributions:**

This paper proposes a class of stochastic gradient coordinate descent-type algorithms with random search directions beyond the standard basis vectors. It presents three types of theoretical results: 1) almost sure convergence; 2) asymptotic normality; and 3) non-asymptotic convergence rates.

**Audience:**

Yes

**Broader Impact Concerns:**

No broader impact concerns.

**Claims And Evidence:**

Yes

**Requested Changes:**

In the end of the proof of Theorem 1, it is said the proof follows that of Theorem 1 in (Bercu et al., 2024)." Can the authors provide a self-contained proof and also acknowledge which part is an analogue of (Bercu et al., 2024)?

Please also explain the last step in the proof of Theorem 2. What is used from Theorem 2.3 in (Zhang, 2016)?

The same request applies to other theorems in the paper. It would be helpful if the authors could state the technical results from other papers in the appendix. Then the paper would be more readable.

**Strengths And Weaknesses:**

The paper provides a complete study of the proposed class of algorithms from almost sure convergence to asymptotic normality and non-asymptotic convergence rates. It also presents numerical experiments on a logistic regression problem to illustrate the effectiveness of the algorithms.

There are two obvious shortcomings of the paper.

First, the paper heavily relies on other papers, for example, (Bercu et al., 2024) and (Chen et al., 2024). The idea of using different directions sampled from underlying distributions is from (Chen et al., 2024). The choices of direction distributions were proposed in Subsection 3.1 of (Chen et al., 2024). In addition, many proofs in this paper rely on results from (Bercu et al., 2024) and other related papers. See the detailed comment in the "Requested Changes." This severely undermines the contribution of this paper.

Second, from all figures and tables presented in Section 5, it seems that the choice (U), i.e., uniform in the canonical basis, is the best among all choices of direction distributions. This is the well-studied and very common choice. This greatly diminishes the relevance of alternative direction distributions, since in practice they offer no advantage over SGD in terms of performance and running time.

---

> ### Author Response · Authors · 2025-06-13
>
> We sincerely thank you for your constructive comments that helped to improve the paper substantially. Our answers are as follows :
>
> > 1. In the end of the proof of Theorem 1, it is said the proof follows that of Theorem 1 in (Bercu et al., 2024)." Can the authors provide a self-contained proof and also acknowledge which part is an analogue of (Bercu et al., 2024)?
>
> The end of the proof of Theorem 1 consists of applying the Robbins-Siegmund theorem on inequality (24) as in the proof of Theorem 1 of Bercu et al.(2024). As you requested, we have now added the full end of the proof for more clarity.
>
> > 2. Please also explain the last step in the proof of Theorem 2. What is used from Theorem 2.3 in (Zhang, 2016)?
>
> We used Theorem 2.3 of Zhang (2016) which is a general central limit theorem for stochastic algorithms. In the proof, we have shown that the two hypothesis needed to use it are satisfied. On the one hand, it is necessary to check that
>
> $$
> \lim_{n\to +\infty} \dfrac{1}{n} \sum_{k=1}^n \mathbb{E} [ \varepsilon_k \varepsilon_k^T \lvert \mathcal{F}_{k-1} ] = \Gamma \qquad a.s.
> $$
>
> One the other hand, we also have to prove that the well-known Lindeberg's condition is satisfied: that for all $\epsilon >0$
>
> $$
> \lim_{n\to +\infty}\dfrac{1}{n}\sum_{k=1}^n \mathbb{E} \left[\lVert\varepsilon_k\rVert^2 I_{\{ \lVert\varepsilon_k\rVert \geqslant \epsilon \sqrt{n}\}}\lvert \mathcal{F}_{k-1} \right]= 0 \qquad a.s.
> $$
>
> Thus, our proof consists in showing that these two conditions are verified for the class of stochastic gradient descent algorithms with random search directions and then applying Theorem 2.3 of Zhang (2016, page 7).
>
> > 3. The same request applies to other theorems in the paper. It would be helpful if the authors could state the technical results from other papers in the appendix. Then the paper would be more readable.
>
> We have taken your request into account and we have now added one Lemma A.3. from Bercu and Bigot (2021) in the appendix.

---

### Comment · Action_Editor_TxWP · 2025-07-16
**Situating your results with respect to the literature**

Dear Authors,

I'm very sorry for engaging so late in the game. Given the reviewers find the theoretical results are novel, sufficiently clear, and the claims match the contributions, I am considering accepting this paper for TMLR. But I have a just few questions and remarks regarding situating your results within the literature.
Essentially I would like to see just some comments as to how your assumptions, and convergences results compare to what is known about SGD and projected versions of SGD. Reviewers please feel free to chime in here.

1. First,  would you mind please commenting on how your results compare to this 2018 Neurips paper:

F. Hanzely, K. Mishchenko and P, Richt arik, *SEGA: Variance Reduction via Gradient Sketching*, Neurips 2018

and this 2019 paper:

David Kozak, Stephen Becker, Alireza Doostan, Luis Tenorio, *Stochastic subspace Descent*, https://arxiv.org/pdf/1904.01145

In both of these papers they have a slightly generalized version of your method given by
$$ x_{t+1} =x_t - \gamma_t P_t P_t^\top \nabla f(x_t) $$
where $P_t$ is a random matrix such that $\mathbb{E}[P_tP_t^\top] =I$.
The difference here being that the they use the full batch gradient $\nabla f(x_t).$ But excluding this, many of the assumptions are the same, and they establish the same type of asymptotic convergence of the iterates (a.s) and sub optimality. This paper is perhaps the most closely related work to yours. In light of this work, and any follow-up work, can you claim you present a "new class of stochastic gradient descent algorithms"?

2. Non-asymptotic results + Restricted Secant Inequality:

Your assumption 6 is (sometimes) known as the RSI (Restricted Secant Inequality), see for instance:

Karimi, H., Nutini, J., & Schmidt, M. (2016). *Linear convergence of gradient and proximal-gradient methods under the Polyak-Łojasiewicz condition*. In *Machine Learning and Knowledge Discovery in Databases – ECML PKDD 2016* (pp. 795–811). Springer.

In particular in Theorem 2, they show that if the gradient is Lipschitz (Your assumption 4 or 7 with p=1) then RSI implies the Polyak-Łojasiewicz (PL) condition, thus it is stronger than the PL condition, and your results could be compared to the convergence of SGD in the smooth +PL setting. For instance see Theorem 5.10 in:

G. Garrigos, R. M. Gower, *Handbook of Convergence Theorems for (Stochastic) Gradient Methods*, arXiv:2301.11235


3. Non-asymptotic results + weak Restricted Secant Inequality:

I have not seen results using your Assumption 3, that is the Restricted Secant Inequality with $\mu =0$. But again from Theorem 2 in

Karimi, H., Nutini, J., & Schmidt, M. (2016). *Linear convergence of gradient and proximal-gradient methods under the Polyak-Łojasiewicz condition*. In *Machine Learning and Knowledge Discovery in Databases – ECML PKDD 2016* (pp. 795–811). Springer.

we have that for smooth functions (your assumption 4 or 7 with p=1) that Restricted Secant Inequality with $\mu =0$ is a slightly weaker assumption than Weak strong convexity with $\mu =0$ that is

$$ f(x_*) \geq f(x) + \langle \nabla f(x), x_*-x \rangle $$

which is known as star-convexity. And there are results for smooth + star convex functions for SGD,

R, M. Gower, O. Sebbouh & N. Loizou *SGD for Structured Nonconvex Functions: Learning Rates, Minibatching and Interpolation*, AISTATS 2021.


4. Using prior results (Bercu et. al 2024):

Using prior results from Bercu et .al 2024 is absolutely fine, but you must state exactly the result you are using. Concluding a proof with "The proof of theorem is completed by proceeding as in the proof of Theorem 1 in (Bercu et al., 2024)" passes the onus to the reader to figure out how to use the techniques in Theorem 1 in (Bercu et al., 2024) to conclude the proof. Instead, you need to state exactly what result/technique/step you need from Bercu et .al 2024 to conclude the proof.

---

> ### Author Response · Authors · 2025-07-21
>
> We would like to thank the action editor for the valuable feedback and the constructive comments. We provide detailed response to each point below.
>
>
> 1.
>
> The methods proposed by Hanzely et al. (2018) and Kozak et al.(2019) are very similar to our SCORS algorithm. However, as you mentioned, the significant difference with us is that we use random estimate of $\nabla f(X_n)$ uniformly picked at each iteration, and not the full gradient.
>
> Hanzely et al.(2018) established the convergence only in expectation by assuming that $f$ is $\mu$-strongly convex. We also highlight that Kozak et al.(2019) proved the almost sure convergence of $(X_n)$ under the same restrictive assumption of strong-convexity. However, our analysis to obtain the almost convergence of the iterates $(X_n)$ just required Assumption 3 which is a weaker condition. This assumption can be seen as a local strict convexity of the function $f$.
>
> Moreover, our other contribution compared to both of these papers, lies in the use of the decreasing step $\gamma_n$ which doesn't depend on $\mu$-strong convexity and $L$-smoothness constants.
> Finally, we shown the central limit theorem for stochastic coordinate gradient descent algorithm which is a novel result.
>
> Given this previous work, we agree that the claim "new class of stochastic gradient descent algorithms" is no longer very appropriate. Thus, we have now made this correction in our article. Moreover, we have chosen to cite these papers in order to present a more complete state of the art.
>
>
> 2.
>
> We are not unfortunately able to compare directly our result to those of Garrigos and Gower (2024).
> Actually, Theorem 5.10 in (Garrigos and Gower, 2024) provides the non-asymptotic rate of convergence on $\mathbb{E}\big[ f(X_n) - f( x^* )\big]$. However, the PL condition is not sufficient to obtain the bound of $\mathbb{E}\big[\lVert X_n - x^* \rVert^2\big]$. In this case, we propose to use the RSI as minimal condition. Moreover, this non-asymptotic rate of convergence on $\mathbb{E}\big[\lVert X_n - x^* \rVert^2\big]$ is a stronger result because it implies the one on $\mathbb{E}\big[ f(X_n) - f( x^* )\big]$ when the function $f$ is $L$-smooth by using the inequality
>
> $$ f(X_n) - f( x^* ) \leqslant \dfrac{L}{2} \lVert X_n-x^* \rVert^2. $$
>
> Finally, remark also that Theorem 5.10 is valid only for a constant stepsize which significantly improves the convergence rate when it is chosen appropriately.
>
>
>
> 3.
>
> Gower et al.(2021) established very interesting results on $\mathbb{E}\big[ f(X_n)-f(x^*)\big]$ for the SGD algorithm. However, we did not use Assumption 3 to obtain the non-asymptotic results. Our analysis required this condition only to provide the almost sure convergence of the iterates $(X_n)$. We replaced Assumption 3 by Assumption 6 which is the RSI in order to establish the non-asymptotic bounds. In that framework, the same arguments as in the previous point 2) remain valid.
>
>
> 4.
>
> We sincerely thank once again the action editor for the helpful suggestions. As you requested, we have now added the full end of the proof to make the techniques used more clear.
>
> In the new version of our article, we have included the discussions on assumptions. Moreover, we have cited these papers in order to have a more complete state of the art.

---

> > ### Comment · Action_Editor_TxWP · 2025-07-22
> > **Include this further context, and all good**
> >
> > Dear authors,
> >
> > Thank you for looking into all of my questions, and addressing each one. I will now recommend that the paper be accepted, but conditioned on the minor  changes we discussed here. I'll send some more details in my my final decision messages.
> >
> > Regards, AC

---

### Decision · Action_Editor_TxWP · 2025-07-22

**Recommendation:** Accept with minor revision

**Additional Comments:**

The paper delivers a comprehensive study on the convergence of the randomly projected stochastic gradient method. Some of the techniques are also interesting and may find applications elsewhere. My acceptance however is conditioned that the following changes be committed:

1) That you add Hanzely et al. (2018) and Kozak et al.(2019) to the background, contextualizing what your paper brings in addition

2) Add context on the Restricted Secant Inequality, perhaps mentioning it's relation to Polyak-Łojasiewicz (PL) condition and/or  star convexity. Here you can take liberty to add what you find is most relevant, and pertinent for your paper.

3) State exactly what result/technique/step you need from Bercu et .al 2024 to conclude the proof.

4) [Optional]  Could you consider discussing one more application of your work to fine-tuning. Since you need only to compute $v^\top g\_t$, where $g\_t$ is a stochastic gradient, you can compute this using a forward pass over the model (or network). Forward passes  for a directional derivative use far less memory than backward passes. This opens up applications in fine-tuning, since the bottleneck in fine-tuning is often memory. See for instance the reference:

 "Fine-Tuning Language Models with Just Forward Passes",
Sadhika Malladi, Tianyu Gao, Eshaan Nichani, Alex Damian, Jason D. Lee, Danqi Chen, Sanjeev Arora, Neurips 2023

This paper focuses on zero order methods, but their motivation would work equally well for computing directional derivatives.  Discussing this application could broaden the reach of your work.

**Audience:**

Yes

**Audience Explanation:**

The paper analyses a class of randomly projects stochastic gradient descent. This method may find applications in fine-tuning, since the projected gradient can be computed in forward-mode with a low memory footprint.

**Claims And Evidence:**

Yes

**Claims Explanation:**

The paper is clear on it's theoretical contributions, and after the adjustments due to the review, the proofs can be easily verified.

---

> ### Author Response · Authors · 2025-07-28
> **Camera Ready Revision**
>
> Dear Action Editor,
>
> We are very pleased to hear that the paper has been accepted. Once again, we would like to thank both the reviewers and AE for the valuable feedback and thoughtful discussions that have helped improve the quality and clarity of the manuscript.
>
> We have incorporated the revisions suggested by the AE and we have now uploaded the final camera ready version of the manuscript.